# Preparation of Polyvinyl Imine Modified Carbon Quantum Dots and Their Application in Methotrexate Detection

**DOI:** 10.3390/molecules27165254

**Published:** 2022-08-17

**Authors:** Xiaojing Si, Mei Han, Hongyan Zeng, Xiaoyi Wei

**Affiliations:** 1Department of Food Science, Shanghai Business School, Shanghai 200235, China; 2School of Materials and Chemical Engineering, Pingxiang University, Pingxiang 337055, China

**Keywords:** polyvinyl imine, methotrexate, carbon quantum dots, fluorescence

## Abstract

Objective: A sensitive and selective fluorescence-detection platform based on carbon quantum dots (CQDs) was designed and developed for the determination of methotrexate (MTX), for the purpose of minimizing the possible toxic threat of MTX in clinics. Methods: The approach was prepared for the first time by a simple, hydrothermal method, making the synthesis and modification processes realized in one step using polyethyleneimine (PEI), and the proposed PEI-CQDs were obtained with high fluorescence quantum yield (38%). Results: MTX was found highly responsive and effective in quenching the fluorescence of the PEI-CQDs, due to a suggested fluorescence resonance energy transfer mechanism or inner-filter effect. The linear range of MTX was between 1 and 600 μmol/L under optimum conditions, with a detection limit (LOD) as low as 0.33 μmol/L. Furthermore, the fluorescent method was established for the MTX assay, and satisfactory results were acquired in real-sample determination. The average labeled quantity was 98.2%, and the average added standard recovery was 100.9%. Conclusions: The proposed PEI-CQDs showed a remarkable potential for broad applications in biological molecule determination and environmental analysis.

## 1. Introduction

Currently, methotrexate (MTX) is a commonly used anti-folic acid and anti-tumor drug, mainly applicable to leukemia meningeal spinal infiltration, acute leukemia, head and neck tumors, bone tumors, breast cancer, liver cancer, lung cancer, and other diseases [1,2]. MTX has been identified as a folate antagonist capable of inhibiting dihydrofolate reductase, lowering DNA synthesis, and reducing cell proliferation and mitosis [3,4]. As an antineoplastic or cytotoxic agent, MTX also targets normal cells and tissues, which may result in alopecia, suppression of the bone marrow, and potentially fatal cirrhosis and pneumonitis [5,6,7]. Due to the increasing use and range of methotrexate drugs, monitoring the MTX level in biological samples is important for studies of efficacy, dosing schedules, different individual responses related to genetic polymorphism, and adverse drug reactions. Moreover, MTX contamination has been discovered in hospitals and sewage-treatment-plant effluent [8]. The presence of chemotherapeutic agents in the environment causes a possible detrimental impact on human health and ecological safety. As a result, contamination with chemotherapeutic agents should be monitored in both biological and environmental samples. Therefore, it is urgent to establish a rapid detection method for MTX.

At present, among various detection techniques applied to MTX, for example, are electrochemical [9,10,11], liquid chromatography [12,13,14], surface plasmon resonance (SPR) [15], and immunoassays [16]. The fluorescence approach has several advantages, including high sensitivity, quick response, and low cost, and the approach was successfully applied for the detection of MTX in clinical samples.

Carbon quantum dots (CQDs) are a new class of carbon nanomaterials. They are quasi-spherical carbon nanoparticles, with a core–shell structure and a size of less than 10 nm. They exhibit several remarkable properties, including ease of preparation, non-toxicity, and excellent luminescence behavior [17,18,19]. CQDs are primarily composed of C, H, and O elements, with amorphous carbon or nanocrystal carbon serving as the core. Diamond-like structures formed by sp3 hybridization have also been observed. They were first obtained in 2004, during the preparative electrophoresis purification of single-walled carbon nanotubes, and then in 2006, during the laser ablation of graphite powder and cement [20]. Additionally, the surfaces of CQDs are rich in carboxyl, hydroxyl, and other functional groups, which imparts strong water solubility and facilitates surface functionalization or passivation with polymers, biomaterials, or inorganic materials. Meanwhile, in order to improve the fluorescence properties of CQDs, heteroatom doping has often been proposed, due to the fact that it can effectively adjust the intrinsic structure and electronic distribution of CQDs, improve quantum yield, and be beneficial to applications [21,22]. Surface modification is an important issue, regardless of future CQD applications, because it can improve quantum yield, enhance photostability, or allow interaction with certain drugs, proteins, biomolecules, cells, and tissues [23]. Therefore, the application of CQDs as fluorescent probes makes great achievements in the fields of fluorescence sensing, photocatalysis and photochromism, and metal-ion detection. Moreover, CQDs also bring important advances to a new level in the fields of biomedicine, in vivo biological imaging, and drug delivery, due to its good biocompatibility and low cytotoxicity [24].

Polyethylene imine (PEI) is a type of amphiphilic cationic polymer (Figure 1). Due to the abundant primary amine, secondary amine, and tertiary amine in the molecular structure, PEI has good solubility in water. The abundant amino also provides a wide range of reaction sites. The hydrophobic groups on the surface and the hydrophobic-layer interaction of CQDs, as well as the hydrophilic layer exposed on the outside, protect the internal structure of the CQDs [20,25]. Furthermore, it has improved its water solubility and stability while retaining the stable and unique optical properties of CQDs. Meanwhile, abundant functional groups and double bonds provide a good foundation for the surface modification of CQDs. Subsequently, CQDs can be modified by surface passivation or doping heteroatoms to show strong luminescence properties, considerably improving their physical and chemical properties [26].

In conclusion, we propose a novel fluorescence probe based on PEI-CQDs for the fast and selective determination of MTX. The obtained fluorescence probes were prepared by a simple, hydrothermal method, making the synthesis and modification processes realized in one step. Moreover, the fluorescence quantum yield (QY) of the PEI-CQDs was 38%, and the PEI-CDs were used as the fluorescent probes for the determination of MTX in the phosphate buffer solution (PBS) and even in human serum, exhibiting not only good sensitivity but also satisfactory practicability.

## 2. Experimental Section

### 2.1. Instrumentation

Transmission electron microscopy (TEM) (JEOL, JEM-2100) with a 200 kV operating voltage was used to characterize the microstructure and the sizes of the PEI-CQDs. Fourier-transform infrared (FTIR) spectra of pressed KBr pellets were recorded using an AVATAR-370 spectrometer (Nicolet, WI, USA) over the spectral range 400–4000 cm^−1^. The surface and functional properties of PEI-CQDs were characterized using an X-ray diffractometer (XRD) equipped with a D8 advance (Bruker, Karlsruhe, Germany). The photo-luminescence (PL) spectra of the samples were recorded on a UV-2501PC spectrophotometer (Shimadzu, Kyoto, Japan) and an F-7000 fluorescence spectrophotometer (Hitachi, Tokyo, Japan), respectively. 

### 2.2. Materials

Methotrexate hydrate (>97%) was obtained from Shanghai Titan Technology Co., Ltd. (Shanghai, China). Polyethyleneimine (>99%, MW 1800) was purchased from Shanghai Vokai Chemical Reagent Co., Ltd. (Shanghai, China). Sinopharm Chemical Reagent Co., Ltd. (Shanghai, China) provided the ethanediol (>99.5%) and the remaining reagents used in this work. Shanghai Xinyi Pharmaceutical Co., Ltd. (Shanghai, China) produced the methotrexate tablets. All analytical measurements were performed using ultrapure water (18.25 MΩ cm).

### 2.3. Synthesis of the PEI-CQDs

PEI-CQDs were synthesized by a hydrothermal technique that utilized glycol as a source of carbon and PEI as a passivating agent for the surface. This approach was previously described in another study, with slight modifications [27]. First, 10 mL of glycol and 2 mL of ultra-pure water were mixed with an ultrasound to form a clear solution. Then, 0.54 g of PEI and 3.5 mL of 85% phosphoric acid were added to the mixed solution, and the solution was magnetically stirred at room temperature for 30 min. Finally, the mixture was added to a high-pressure, nitrifying pot (30 mL) and heated at 180 °C in a constant-temperature drying oven for 24 h to form PEI-CDs. The obtained brown liquid was centrifuged for 15 min at 5000× *g* rpm to remove large carbon particles and unreacted substances, and the upper solution was filtered with a 0.22 µm filter membrane, with the constant volume at 50 mL, and then stored away from light for later use.

### 2.4. Determination of Quantum Yield

The fluorescence quantum yield (*QY*) of PEI-CQDs can be determined and calculated by the following equation [28].
(1)$$QY=QY_{R}\times \frac{I}{I_{R}}\times \frac{A_{R}}{A}\times \frac{n^{2}}{n_{R}{}^{2}}$$
where quinine sulfate is used in the present work, and *QY_R_* = 54%. *I* and *I_R_* are the measured, integrated emission intensity of PEI-CQDs and quinine sulfate, respectively. *A* and *A_R_* represent the absorbance of PEI-CQDs and quinine sulfate, which are measured on a UV–Vis spectrophotometer. The quinine sulfate was dissolved in 0.1 M of H_2_SO_4_ (*n* = 1.428), and the PEI–CQDs were dissolved in distilled water (*n* = 1.333).

### 2.5. Pretreatment and Detection of MTX Tablets

A total of 10 MTX tablets were weighed and fully ground. Then, the powder was extracted with 0.1 mol/L of NaOH solution for 30 min in an ultrasonic bath until sufficiently dissolved, transferred to a clean, 100 mL volumetric flask for constant volume, and mixed well. The solution was diluted to the exact volume with distilled water, and then filtered by a 0.22 μm filter membrane. At the end, the sample solution was 1 mg/mL, and it was stored in the dark and was used as the real sample’s solution.

The following procedure was used to determine the presence of MTX in real samples: 250 μL of PEI-CQDs solution was poured into a series of colorimetric tubes (10 mL), and then 0.01 mol/L of MTX solution in different volumes was added to each colorimetric tube in turn, followed by the phosphate buffer solution (PBS) (pH 7.0) as the calibration line. After 3 min, the fluorescence intensity was detected at an excitation wavelength of 350 nm.

## 3. Results and Discussion

### 3.1. Structural Analysis of PEI-CQDs

During the synthesis of PEI-CQDs, the amine group in dendritic PEI can react with the hydroxyl group in glycol to form a hydrogen bond and coat the carbon point. The carbon elements on the surface of the fluorescent carbon spots form carboxyl groups under the oxidation of concentrated phosphoric acid, making them hydrophilic and providing active groups for subsequent biological applications to improve luminescence performance. The QY of PEI-CQDs was calculated using the method described, and the result was 38%, according to Section 2.4. The TEM images in Figure 2A clearly show the resultant PEI-CQDs, which have a well-defined, hyperbranched structure and a uniform particle-size distribution, with a diameter ranging between 3 and 7 nm and a polydispersity index (PDI) of 0.383, as shown in Figure 2B. CQDs have been widely reported to generate photoluminescence as a result of their band-gap transition and electron-hole recombination caused by their surface defects [29]. XRD was used to investigate the crystallinity and crystal structure of PEI-CQDs, and the resulting XRD pattern is shown in Figure 2C. The diffraction peak of amorphous carbon was observed at 2θ = 22.52°, which was consistent with the (100) lattice spacing of the carbon-based materials, and the crystal structure of PEI-CQDs was determined to be amorphous carbon [30].

The chemical composition and functional groups on the surface of the developed PEI-CQDs were investigated. The FT-IR spectrum of the synthesized PEI-CQDs is shown in Figure 3. The broad absorption band between 3500 and 3000 cm^−1^ is attributed to N-H and O-H stretching vibrations, and the absorption peak is stronger than those of ethylene glycol and PEI, showing the presence of amino and hydroxyl groups on the surface of the PEI-CQDs, which contributes to their hydrophilic properties. The peaks at 1653 cm^−1^ and 1380 cm^−1^ corresponded to the NH/NH_2_ N-H stretching vibration and the C-H stretching vibration, respectively. The absorption peak at 1095 cm^−1^ is attributed to the stretching vibration of C-O-C [31].

### 3.2. Optical Properties

UV-vis absorption and photoluminescence-emission spectroscopy were used to characterize the optical properties of PEI-CQDs. As illustrated in Figure 4, the UV–vis absorption spectrum (a) of PEI-CQDs exhibits a characteristic peak at 320 nm corresponding to the C=O *n*−π * transition, while PEI have nearly no absorption above 240 nm, which was a difference between PEI-CQDs and PEI [26]. Meanwhile, a strong emission peak at 460 nm (c) appeared upon excitation at 350 nm (b), which may be the result of the surface states trapping excited-state energy. All in all, the wavelengths of excitation and emission for the following experiments were set at 350 nm and 460 nm, respectively.

### 3.3. Analytical Performance of the PEI-CQDs for MTX

Several related factors, such as the pH and the volume of the PEI-CQDs solution, were investigated. As illustrated in Figure 5A, the fluorescence quenching of PEI-CQDs was pH-dependent and relatively low when 50 μL of 0.01 mol/L MTX was used in 10 mL of PBS under neutral and alkaline conditions. In comparison, the quenching of fluorescence by PEI-CQDs remained stable in neutral solutions with a pH of 7.0. Therefore, MTX was measured in a PBS solution with a pH of 7.0 because it is well-studied in vivo. As we know, MTX can quench the fluorescence of PEI-CQDs. In this work, the fluorescence intensity of PEI-CQDs without MTX was denoted as F_0_, and F was the fluorescence intensity when MTX was added. In Figure 5B, the fluorescence intensity was lower in the presence of 50 μM MTX than in the absence of it, regardless of the amount of PEI-CQDs added. The quenching effect was the most significant when the volume of PEI-CQDs was 250 μL because the maximum ΔF (ΔF = F_0_ − F) was obtained.

To assess the analytical performance, MTX samples at various concentrations were mixed with PEI-CQDs under optimal conditions. The fluorescence quenching of PEI-CQDs toward MTX appeared at concentrations ranging from 1 to 600 μmol/L, as shown in Figure 6.

This result is consistent with the equation log (F_0_/F) = 0.00284 C_MTX_ + 0.04575, which has a correlation coefficient of R^2^ = 0.9976. At a signal/noise ratio of 3, the limit of detection (LOD) is 0.33 μmol/L and the limit of quantitation (LOQ) is 1.0 μmol/L. This fluorescent method, as developed, can successfully be used to monitor MTX in real time. The presented method can provide a wide linear range with other previous methods (shown in Table 1).

### 3.4. The Reaction Mechanism of PEI- CQDs with MTX

MTX greatly quenches the fluorescence intensity of the PEI-CDs with the increase in the concentration of MTX, as shown in Figure 6A. Additionally, the mechanism by which the PEI-CQDs quench fluorescence was investigated. According to previous research, the primary quenching mechanisms for quenching fluorescence carbon dots are dynamic quenching (DQ), static quenching (SQ), photoinduced electron transfer (PET), inner-filter effect (IFE), and fluorescence resonance energy transfer (FRET) [37,38]. Carbon quantum dots are excellent electron acceptors and donors [39]. Because there are aromatic ring functional groups in the structure of methotrexate, and PEI-CQDs show that there are a large number of -NH- and -NH_2_ functional groups, the reason for fluorescence quenching is due to PET, the electron transfer process between photoexcited PEI-CQDs, and the aromatic functional groups of MTX.

### 3.5. Selectivity of the PEI-CQDs for MTX Detection

The complexity of biological and environmental samples makes detecting chemotherapeutic drugs extremely difficult. Thus, experiments on selectivity were conducted in this work. The fluorescence intensities of PEI-CQDs were determined in the presence of a series of drugs and a group of potentially interfering counterparts. Some chemicals were added to the fluorescence-intensity-detection system (50 μL of 0.01 mol/L MTX), and the results are shown in Table 2. Most chemical compounds can coexist at higher concentrations with a relative error of less than 10%.

### 3.6. MTX Detection in Real Samples

The developed method’s applicability was evaluated using MTX tablets purchased from a local drug store. The standard addition method was performed in human serum provided by different healthy volunteers from the infirmary of Shanghai University. The 100 μL human serum was diluted 100 times with PBS (pH 7.0), and then 100 μL supernatant containing MTX (1 mg/mL) was added. The average labeled quantity was 98.2%, with a relative standard deviation (RSD) of 4.4% (Table 3). Furthermore, the recoveries obtained from the samples spiked with standard MTX solutions range from 99.7% to 101.0%. Each sample had an RSD of 0.9%, indicating good reproducibility and precision (Table 4). As a result, the proposed method is found to be feasible for MTX detection in drug samples.

## 4. Conclusions

In conclusion, PEI as a surface passivation agent that modified CQDs was successfully prepared using a one-step hydrothermal method with excellent water solubility and higher quantum yield. The conditions were optimized. The fluorescence quenching of PEI-CQDs by MTX was caused primarily by an inner-filter effect based on the agglomeration and electron transfer of PEI-CQDs. The as-prepared PEI-CQDs demonstrated excellent MTX-detection performance, with linear dynamic ranges ranging from 1 to 600 μmol/L and a detection limit of 0.33 μmol/L. Furthermore, the MTX spiked in drug samples monitored by the method recovered well. We anticipate that this approach can be used for developing more sensitive sensing methods in drug analysis and environmental monitoring.

## Figures and Tables

**Figure 1 molecules-27-05254-f001:**
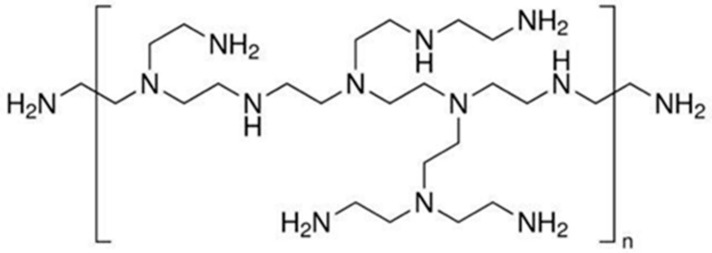
The structure of branched PEI.

**Figure 2 molecules-27-05254-f002:**
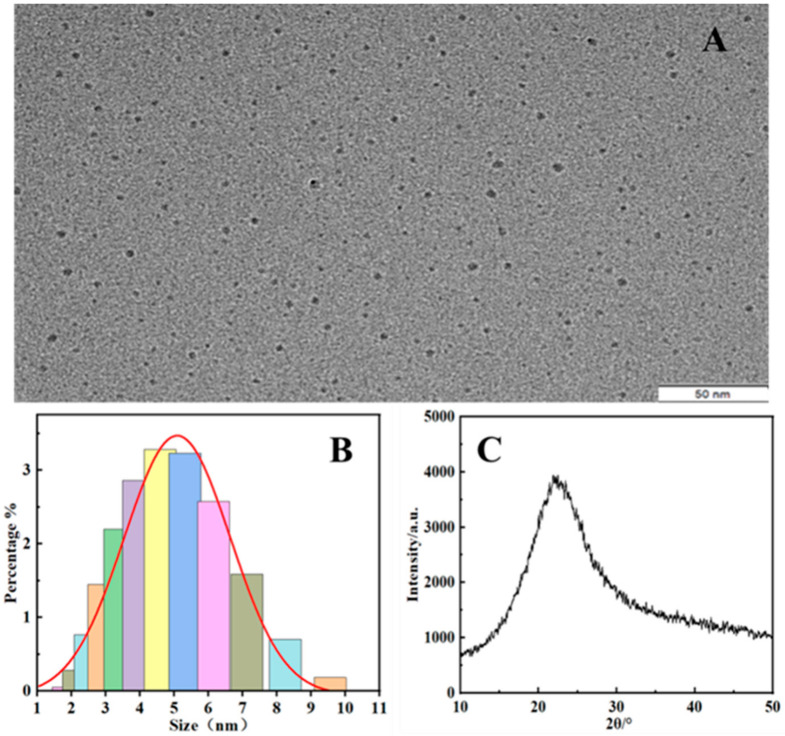
(**A**) TEM image of the synthesized PEI-CQDs; (**B**) PEI-CQDs’ particle-size distribution; (**C**) XRD pattern of PEI-CQDs.

**Figure 3 molecules-27-05254-f003:**
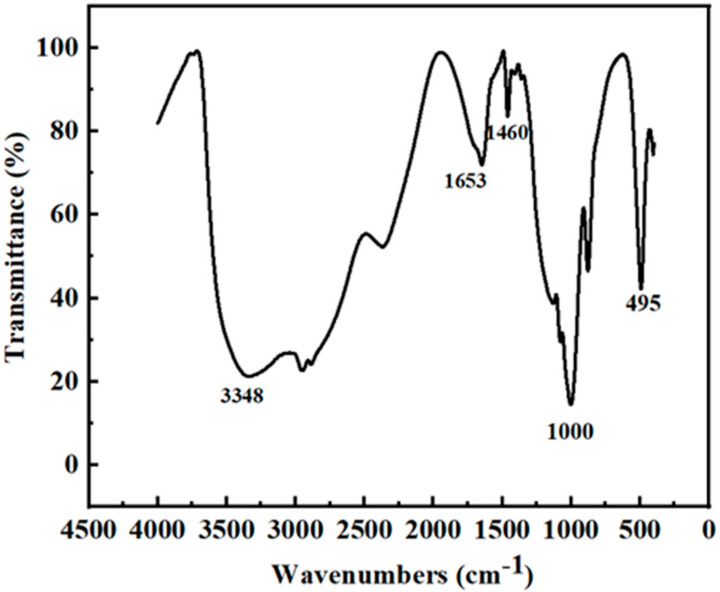
FT-IR spectrum of the PEI-CQDs.

**Figure 4 molecules-27-05254-f004:**
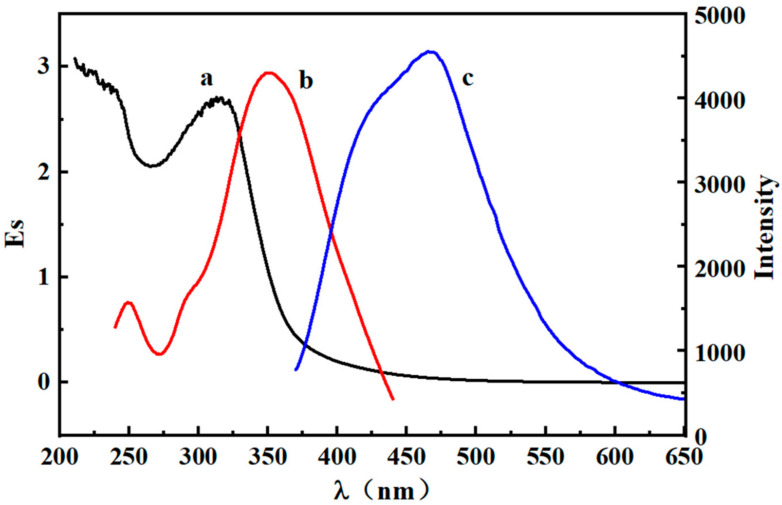
UV−vis and PL absorption spectra of PEI-CQDs.

**Figure 5 molecules-27-05254-f005:**
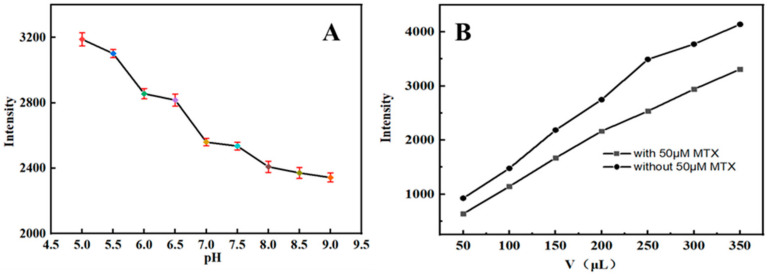
Effect of pH (**A**) and volume (**B**) of PEI-CQDs on the fluorescence intensity of the detection system.

**Figure 6 molecules-27-05254-f006:**
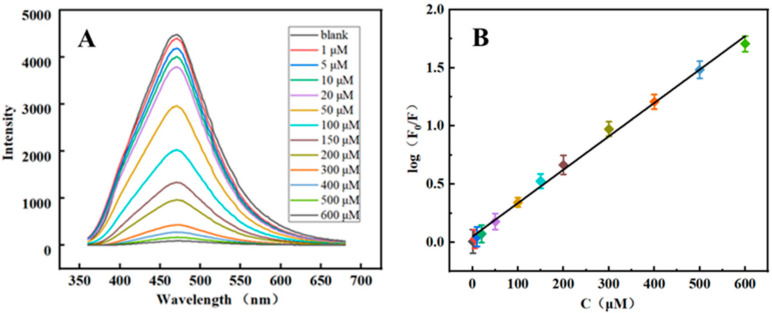
(**A**) Fluorescence-intensity curves of MTX with different concentrations; (**B**) Linear relation curve of log (F_0_/F) and concentration of MTX.

**Table 1 molecules-27-05254-t001:** Comparison of major characteristics for MTX with other published methods.

Method	Linear Range	Detection Limit	References
N,S-CQDs	0.4–41.3 μg/mL	12 ng/mL	[32]
CE	0.5–10 μmol/L	0.1 μmol/L	[33]
SPRS	0–150 μmol/L	0.6 μmol/L	[15]
UPLC-MS-MS	1.0–100μmol/L	0.16μmol/L	[34]
chemiluminescence immunoassay	4.3–392.8 ng/mL	9.1 ng/mL	[16]
Electrochemical sensor	0.005–7 μmol/L	3.07 nmol/L	[35]
Flow-injection–electrochemical oxidation fluorimetry	2.0 × 10^−7^–1.0 × 10^−5^ g/mL	5.2 × 10^−8^ g/mL	[36]
PEI-CQDs	1–600 μmol/L	0.33 μmol/L	This work

**Table 2 molecules-27-05254-t002:** Interference of different substances on the determination of methotrexate.

Interfering Substance	Concentration (μmol/L)	Current Ratio %
NaCl	2500	98.54
KNO_3_	5000	100.81
CH_3_COONH_4_	5000	100.81
K_2_SO_4_	2500	100.31
Glycine	2500	101.74
Phenylalanine	2500	102.52
Histidine	2500	103.18
Lysine	2500	104.33
Ascorbic acid	5000	101.99

**Table 3 molecules-27-05254-t003:** Determination results of MTX in the pharmaceutical product (*n* = 5).

No.	Detected (μmol/L)	Detected (mg/Piece)	Labeled (mg/Piece)	Labeled Quantity (%)	Average (%)	RSD (%)
1	5.86	2.44	2.5	97.6	98.2	4.4
2	6.04	2.51		100.4		
3	6.23	2.59		103.6		
4	5.52	2.30		92.0		
5	5.83	2.43		97.2		

**Table 4 molecules-27-05254-t004:** MTX standard recovery (*n* = 5).

No.	Sample (μmol/L)	Added (μmol/L)	Total (μmol/L)	Recovery (%)	Average (%)	RSD (%)
1	5.90	100	107.9	102.0	100.9	0.9
2			106.2	100.3		
3			106.8	100.9		
4			107.5	101.6		
5			105.6	99.7		

## Data Availability

The data presented in this study are available on request from the corresponding author. The data are not publicly available due to school rules.

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
