# Peer review of "Preparation of Polyvinyl Imine Modified Carbon Quantum Dots and Their Application in Methotrexate Detection"

_molecules, 2022, doi:10.3390/molecules27165254_

Round 1
Reviewer 1 Report
1. Background content should not appear in the abstract. And it is not appropriate to abbreviate "fluorescent copolymers" as "PEI-CQDs".
2. 2.4 - The processing of drug samples violated the analytical chemistry experiment operating procedures.
3. The description "Fig. 4(A), UV-Vis and PL absorption spectra of PEI-CQDs" is inappropriate. In 3.2, many descriptions are scientifically wrong. As we all know, the fluorescence emission spectrum has nothing to do with the excitation wavelength, and it is meaningless to study the effect of the excitation wavelength change on the fluorescence emission spectrum.
4. The optimization of experimental conditions (pH) is not an analytical characteristic.
5. The mechanism of fluorescence quenching should be briefly discussed.
6. The source of the blood sample is unknown, and the time to collect the blood sample after taking the drug is not specified, and there is a contradiction with the measurement result "mg/piece".
7. F0 and F are not specified, please add explanations.
8. The full text contains a large number of editing errors such as capitalization, subscripts and subscripts.
Author Response
Comment 1. Background content should not appear in the abstract. And it is not appropriate to abbreviate "fluorescent copolymers" as "PEI-CQDs".
Replay: Thank you for your suggestions. The objective has been instead of background in the abstract. The sentences have been added and highlighted in red in the abstract. The PEI-CQDs were synthesized by a hydrothermal technique that utilized glycol as a source of carbon and PEI as a passivating agent for the surface with the phosphoric acid involved, and it is not appropriate to abbreviate "fluorescent copolymers", we have changed it.
Comment 2. 2.4-The processing of drug samples violated the analytical chemistry experiment operating procedures.
Replay: Thank you for your suggestions, it is very useful for us. We have checked the processing of drug samples and modified in section 2.5.
Comment 3. The description "Fig. 4(A), UV-Vis and PL absorption spectra of PEI-CQDs" is inappropriate. In 3.2, many descriptions are scientifically wrong. As we all know, the fluorescence emission spectrum has nothing to do with the excitation wavelength, and it is meaningless to study the effect of the excitation wavelength change on the fluorescence emission spectrum.
Replay: Thank you for your suggestions. For different materials, in most cases, the emission wavelength will shift with the excitation wavelength. For the CQDs solution, the excitation wavelength will also lead to a significant difference in the emission spectrum. In our previous work, which has the same Phenomenon, the emission wavelength and intensity is dependent on the excitation wavelength (Zeng H.Y., Li L., Ding Y.P., Zhuang Q. Simple and selective determination of 6-thioguanine by using polyethylenimine (PEI) functionalized carbon dots. Talanta, 2018, 178, 879-885. DOI:10.1016/j.talanta.2017.09.087). The sentences have been revised and highlighted in red in the section 3.2.
Comment 4. The optimization of experimental conditions (pH) is not an analytical characteristic.
Replay: Thank you for your suggestions. In the work, The fluorescence activity of the PEI–CQDs was found to be pH-dependent. Therefore, in our opinon, it is necessary to determine the optimal pH.
Comment 5. The mechanism of fluorescence quenching should be briefly discussed.
Replay: Thank you for your suggestions. The reaction mechanism of PEI- CQDs with MTX has been added and highlighted in red in the section 3.4.
Comment 6. The source of the blood sample is unknown, and the time to collect the blood sample after taking the drug is not specified, and there is a contradiction with the measurement result "mg/piece".
Replay: Thank you for your suggestions. It is our negligence. The source of the blood sample was from the infirmary of Shanghai University. The sentences have been added and highlighted in red in the section 3.5.
In order to verify the method we developed, two experiments were done.
(1) The content of methotrexate in the actual sample was determined and compared with the labeled amount of the medicine. The result was shown in Table 2
(2) The standard addition method was performed in human serum. The result was shown in Table 3.
I am sorry, the experiment in vivo was not conducted in this work. So, there is no data about the time to collect the blood sample after taking the drug, and we will do further research in the future.
- F0 and F are not specified, please add explanations.
Replay: Thank you for your suggestions. The fluorescence intensity of PEI-CQDs without MTX was denoted as F0, and F was the fluorescence intensity when added MTX. The sentences have been added and highlighted in red in the section 3.3.
- The full text contains a large number of editing errors such as capitalization, subscripts and subscripts.
Replay: Thank you for your suggestions. We have checked and revised one by one.
Reviewer 2 Report
The authors have presented a work of carbon dot nano-assembly from polyvinyl imine for the detection of methotrexate. The manuscript is presented with a fair number of references and have presented the obtained results in a clear manner. However, the reviewer suggests the following revisions to enhance the manuscript quality for accepting for publishing.
1. Authors should present the experimental methods and detection calculation for the concluded quantum yield values in the manuscript.
2. AFM images also should be investigated alongside TEM to support the spherical property of CQDs.
3. From previous literature, CQDs can inherently present excitation dependent emission as well independent. The authors have synthesized PEI-CQDs to excitation independent emission. What are the authors understanding of the PEI-CQDs for this fluorescent property?
4. Polyvinyl imine should be tested as a control blank for the presented results of CQDs- e.g. FT-IR, UV-vis, fluorescence.
5. The possibility of polyvinyl imine quenching as a control should be presented.
Author Response
Comment 1. Authors should present the experimental methods and detection calculation for the concluded quantum yield values in the manuscript.
Replay: Thank you for your suggestions. We have added the experimental methods and detection calculation for the concluded quantum yield in section 2.4 “Determination of quantum yield”.
Comment 2. AFM images also should be investigated alongside TEM to support the spherical property of CQDs.
Replay: Thank you for your suggestions. AFM is a powerful instrument for analyzing the surface structure of solid materials. Now, due to the epidemic and the summer vacation, it is not convenient to go to our university in Shanghai. I am sorry, I can’t finish the experiment in ten days. Fig 2A is the TEM image of the PEI-CQDs, and it is a complete image, we haven't changed it. Therefore, we think the appearance of PEI-CQDs can be clearly characterized in Fig 2A.
Comment 3. From previous literature, CQDs can inherently present excitation dependent emission as well independent. The authors have synthesized PEI-CQDs to excitation independent emission. What are the authors understanding of the PEI-CQDs for this fluorescent property?
Replay: Thank you for your suggestions. For different materials, in some cases, the emission wavelength will shift with the excitation wavelength. The photoluminescence of CDs is that the emission wavelength and intensity is dependent on the excitation wavelength. In this work, because the size effect of PEI-CQDs, the emission wavelengths are red-shifted with the increase of excitation wavelength. And these indicated that the PEI-CQDs have excellent optical performance.
Comment 4. Polyvinyl imine should be tested as a control blank for the presented results of CQDs- e.g. FT-IR, UV-vis, fluorescence.
Replay: Thank you for your suggestions. It is very helpful for us. Compared the FT-IR, UV from previous literatures. Our conclusion is consistent with the literatures.
- Dong Y.Q., Wang R.X., Li H., Shao J.W., Chi Y.W., Lin X.M. and Chen G.N. Polyamine-functionalized carbon quantum dots for chemical sensing. Carbon. 2012, 50, 2810–2815. DOI:10.1016/j.carbon.2012.02.046
- Zeng H.Y., Li L., Ding Y.P., Zhuang Q. Simple and selective determination of 6-thioguanine by using polyethylenimine (PEI) functionalized carbon dots. Talanta, 2018, 178, 879-885. DOI:10.1016/j.talanta.2017.09.087
Comment 5. The possibility of polyvinyl imine quenching as a control should be presented.
Replay: Thank you for your suggestions. The reaction mechanism of PEI- CQDs with MTX has been added and highlighted in red in the section 3.4.
Reviewer 3 Report
The Manuscript title "Preparation of polyvinyl imine modified carbon quantum dots and their application in methotrexate detection" by Xiaoyi Wei et. al reported quantum dot based detection of Methotrexate(MTX) in body fluid using fluorescence based assay. This group reported how quantum dot based approach used to detect toxic material in minute quantity in the sample. They synthesize quantum dot and characterise it for their optimal work. A very novel and excellent work reported by this group which have direct application. This work might be more beneficial for other researcher who working in this area if author provide some more detail as per pointed out quarry mentioned below:
1. In Fig. 1 you mentioned TEM image and particles distribution of these CQD. I am curious about size of these CQD on the basis of TEM image. In particles distribution author should also include PDI value of these CQD. Both data should be incorporated at appropriate place in manuscript.
2. In Fig 3. Author mentioned FT-IR spectra of PEI-CQDs. Author should include spectra of PEI and ethylene glycol. Which will help to add more clarity in this figure and also give confirmation of PEI-CQD synthesis.
Author Response
Response to Reviewer 3 Comments
Comment 1. In Fig. 1 you mentioned TEM image and particles distribution of these CQD. I am curious about size of these CQD on the basis of TEM image. In particles distribution author should also include PDI value of these CQD. Both data should be incorporated at appropriate place in manuscript.
Replay: Thank you for your suggestions. It is our false. We check the origin data and added the PDI in section 3.1. The sentences have been added and highlighted in red in the manuscript.
Comment 2. In Fig 3. Author mentioned FT-IR spectra of PEI-CQDs. Author should include spectra of PEI and ethylene glycol. Which will help to add more clarity in this figure and also give confirmation of PEI-CQD synthesis.
Replay: Thank you for your suggestions. It is very helpful for us. Now, due to the epidemic and the summer vacation, it is not convenient to go to lab in our university in Shanghai.
From previous literature“Zeng H.Y., Li L., Ding Y.P., Zhuang Q. Simple and selective determination of 6-thioguanine by using polyethylenimine (PEI) functionalized carbon dots. Talanta, 2018, 178, 879-885. DOI:10.1016/j.talanta.2017.09.087”,there is a UV FT-IR spectrum of PEI. Zeng is my classmate in Shanghai University, one author of the manuscript, we have compared the spectrums of PEI and PEI-CQDs. It's obviously a big difference.
The FT-IR of ethylene glycol is as follows. There are different among these three spectrums (PEI, PEI-CQDs, glycol) .
Reviewer 4 Report
This work reports on the development of polyvinyl imine-modified carbon quantum dots for the detection of methotrexate. My comments are as follows:
1. Suggest to remove numbering in the abstract section.
2. Suggest to highlight the novelty and main contribution in the abstract and introduction sections.
3. Suggest to replace Fig. 2(A) with a better resolution image since the hyperbranched structure of PEI-CQD is not clearly visible in the present image.
4. Suggest to add FTIR spectrum for CQDs and PEI alone in Fig. 3 to proof that the success of CQDs-PEI preparation.
5. Authors report on the LOD and selectivity of the sensor. How about other sensing performance parameters such as sensitivity, and LOQ?
6. Suggest to compare the sensing performance of this work with other methods to detect methotrexate published in the literature. Summarize it in a table form.
7. There are some units that are not superscript (i.e. cm-1, -1 is not superscripted). Please rechecked throughout the document.
Author Response
Response to Reviewer 4 Comments
Comment 1. Suggest to remove numbering in the abstract section.
Replay: Thank you for your suggestions. We have removed them in the abstract section.
Comment 2. Suggest to highlight the novelty and main contribution in the abstract and introduction sections.
Replay: Thank you for your suggestions. The sentences have been revised and highlighted in red in the abstract and introduction sections.
- Suggest to replace Fig. 2(A) with a better resolution image since the hyperbranched structure of PEI-CQD is not clearly visible in the present image.
Replay:Thank you for your suggestions. Now, due to the epidemic and the summer vacation, it is not convenient to go to our university in Shanghai. Fig 2A is a complete image, we haven't changed it. So, we are sorry for that we cannot do it again in this month.
- Suggest to add FTIR spectrum for CQDs and PEI alone in Fig. 3 to proof that the success of CQDs-PEI preparation.
Replay: Thank you for your suggestions. It is very helpful for us. Now, due to the epidemic and the summer vacation, it is not convenient to go to lab in our university in Shanghai.
From previous literature“Zeng H.Y., Li L., Ding Y.P., Zhuang Q. Simple and selective determination of 6-thioguanine by using polyethylenimine (PEI) functionalized carbon dots. Talanta, 2018, 178, 879-885. DOI:10.1016/j.talanta.2017.09.087”,there is a UV FT-IR spectrum of PEI. Zeng is my classmate in Shanghai University, one author of the manuscript, we have compared the spectrums of PEI and PEI-CQDs. It's obviously a big difference.
The FT-IR of ethylene glycol is as follows. There are different among these three spectrums (PEI, PEI-CQDs, glycol) .
- Authors report on the LOD and selectivity of the sensor. How about other sensing performance parameters such as sensitivity, and LOQ?
Replay: Thank you for your suggestions. The sentences have been added and highlighted in red in section 3.3.
- Suggest to compare the sensing performance of this work with other methods to detect methotrexate published in the literature. Summarize it in a table form.
Replay: Thank you for your suggestions. We have added the table about the Comparison of this work with other methods that published in the literature in section 3.3. (Table.1)
- There are some units that are not superscript (i.e. cm-1, -1 is not superscripted). Please rechecked throughout the document.
Replay: Thank you for your suggestions. It is our false. We have checked and revised one by one.
Round 2
Reviewer 1 Report
I think the author did not understand my real concerns:
1. It is theoretically impossible to transfer the 10 mg tablet powder into a 10 mL volumetric flask after completely dissolving it, and a step-by-step dilution method should be adopted.
2. Figure 4(B) cannot explain that the fluorescence emission spectrum of PEI-CQDs changes with the excitation wavelength. This figure is meaningless and can be deleted.
3. The pH of the solution will affect the emission of PEI-CQDs, its reaction with MTX, etc., it is not an analytical characteristic, but belongs to the category of experimental conditions optimization.
Author Response
1、It is theoretically impossible to transfer the 10 mg tablet powder into a 10 mL volumetric flask after completely dissolving it, and a step-by-step dilution method should be adopted.
Reply:Thank you for your suggestions. It is helpful for us. We have modified the sentence in section 2.5.
2、Figure 4(B) cannot explain that the fluorescence emission spectrum of PEI-CQDs changes with the excitation wavelength. This figure is meaningless and can be deleted.
Reply:Thank you for your suggestions. We complied with your suggestion and deleted the Figure 4(B).
3、The pH of the solution will affect the emission of PEI-CQDs, its reaction with MTX, etc., it is not an analytical characteristic, but belongs to the category of experimental conditions optimization.
Reply:Thank you for your guidance. You are quite right. Through optimizing of pH conditions, 7.0 is the most suitable condition for the determination of MTX. I benefited a lot from these suggestions. Thanks again.
Reviewer 2 Report
The authors have satisfactorily answered majority of the reviewer comments.
Author Response
Thank you for your guidance. I benefited a lot from these suggestions.
We'll work on the English language and style of the manuscript.
Thanks again.